# Unraveling the Seismic Source in Archaeoseismology: A Combined Approach on Local Site Effects and Geochemical Data Integration

**Carla Bottari** [1,*] **, Patrizia Capizzi** [2] **and Francesco Sortino** [3]

1   Istituto Nazionale di Geofisica e Vulcanologia, Sezione di Catania, Osservatorio Etneo, 95125 Catania, Italy
2   Dipartimento di Scienze della Terra e del Mare, Università degli Studi di Palermo, 90123 Palermo, Italy; patrizia.capizzi@unipa.it
3   Istituto Nazionale di Geofisica e Vulcanologia, Sezione di Palermo, 90146 Palermo, Italy; francesco.sortino@ingv.it
*   Correspondence: carla.bottari@ingv.it

**Abstract:** Archaeoseismological research often deals with two unresolved questions: the magnitude and level of damage caused by past earthquakes, and the precise location of the seismic source. We propose a comprehensive review of an integrated approach that combines site effects with the analysis of geochemical data in the field of archaeoseismology. This approach aims to identify active buried faults potentially related to the causative seismic source and provide insights into earthquake parameters. For each integrated method, we report the foundational principles, delineation of theoretical field procedures, and exemplification through two case studies. Site effects analysis in archaeoseismology assumes a pivotal role in unraveling historical seismic occurrences. It enables estimating the earthquake magnitude, assessing the seismotectonic patterns, and determining the resulting damage level. Valuable data related to earthquake parameters can be extracted by analyzing vibration frequencies and acceleration measurements from structures within archaeological sites. This information is instrumental in characterizing seismic events, evaluating their impact on ancient structures, and enhancing our understanding of earthquake hazards within the archaeological context. Geochemical investigations supply indispensable tools for identifying buried active faults. The analysis of fluids and gases vented in proximity to faults yields valuable insights into their nature, activity, and underlying mechanisms. Faults often manifest distinctive geochemical imprints, enabling the differentiation between tectonically active and volcanically related fault systems. The presence of specific gases can further serve as indicators of the environmental conditions surrounding these fault networks. Integrating site effects analysis and geochemical investigations within archaeoseismological research is crucial to improving our understanding of unknown past earthquakes. Moreover, it enhances the seismic hazard assessment of the region under study.

**Keywords:** archaeoseismology; local site effects; earthquake parameters; geochemical investigation; buried active fault

## 1. Introduction

Archaeoseismology, also known as earthquake archaeology, is a specialized field within Earth science that focuses on the documentation of past seismic events. This discipline examines the damage and traces discovered in archaeological sites [1–5], seeking to elucidate sites that are potentially unknown or insufficiently recorded by historical sources, while avoiding circular reasoning [6]. Through a multifaceted approach, archaeoseismologists reconstruct ancient earthquakes' impact on human settlements by analyzing damage patterns, displaced structures, and seismic indicators found in archaeological remains [7–10]. This interdisciplinary approach addresses essential research inquiries, including assessing seismic ground motions' likelihood for having caused the observed

damage, determining the earthquake timing, and inferring the earthquake characteristics [11]. By pursuing these questions, archaeoseismologists aim to comprehend a region's seismic history, offering crucial data for earthquake hazard assessment and insights into past civilizations.

Archaeoseismological field studies adopt an interdisciplinary approach, integrating principles and methods from various disciplines such as archaeology, geology, seismology, geophysics, and engineering. This interdisciplinary emphasis has also been highlighted by Ambraseys [12] and Sintubin [13].

The archaeoseismological methodology is essential for a comprehensive understanding of historical events, especially in periods where the seismic catalogue have scan information, such as the first millennium C.E. Integrating different disciplines facilitates reconstructing historical earthquake impacts, refining the understanding of prolonged seismic dynamics, and contributes significantly to evaluating seismic periods [5,14].

The evolution from qualitative [7,10] to quantitative approaches [8,11,15] in archaeoseismology underscores the need for interdisciplinary collaboration. While qualitative methods involve subjectivity and individual interpretation, quantitative approaches employ analytical and numerical modeling procedures [11]. Quantitative methods yield a more robust assessment of the archaeoseismic damage, but it is crucial to acknowledge the potential subjectivity of qualitative observations influenced by context. Thus, adopting an interdisciplinary approach tailored to the project is recommended for objective damage assessment.

To study the local site effects and identify the causative faults, the integration of geophysical techniques and geochemical data analysis becomes crucial. Archaeoseismologists can combine these methodologies to precisely determine the location of the seismic causative sources, particularly in sites where the evidence is unclear [11,15,16].

Structural deformation examples in ancient buildings, influenced by the local seismic response, persist today. Notably, the Colonna Antonina in Rome experienced significant ground motion due to recent deposits with poor geotechnical characteristics during a historical earthquake [17]. Similar effects were observed in the Temple of Hephaistos and in the Propylaia in Athens [18–20].

Geophysical techniques, such as microzonation, ground motion modeling, and numerical analysis, characterize the seismic behavior of the site [21–23]. Additionally, geochemical analysis involves sampling and laboratory examination of fault-related indicators [24,25]. Integrating these disciplines enables archaeoseismologists to unveil the seismic source, enhancing the understanding of past seismic events and their impact on archaeological sites. Some examples of deformation and earthquake damage are reported in Figure 1.

Summarizing, this paper focuses on a combined approach, using local seismic site analysis and geochemical data integration in archaeoseismology. This interdisciplinary approach enhances the comprehension of past seismic events, particularly those absent from seismic catalogs, contributing valuable data to earthquake hazard assessment and informing contemporary disaster management strategies and urban planning.

The subsequent section delineates the integration of site effect studies and geochemical investigations within the archaeoseismological framework. Each discipline is illustrated by a single case study; for instance, the geophysical survey of Selinunte illustrates the site effect, while the geochemical analysis pertains to the Santa Venera al Pozzo site.

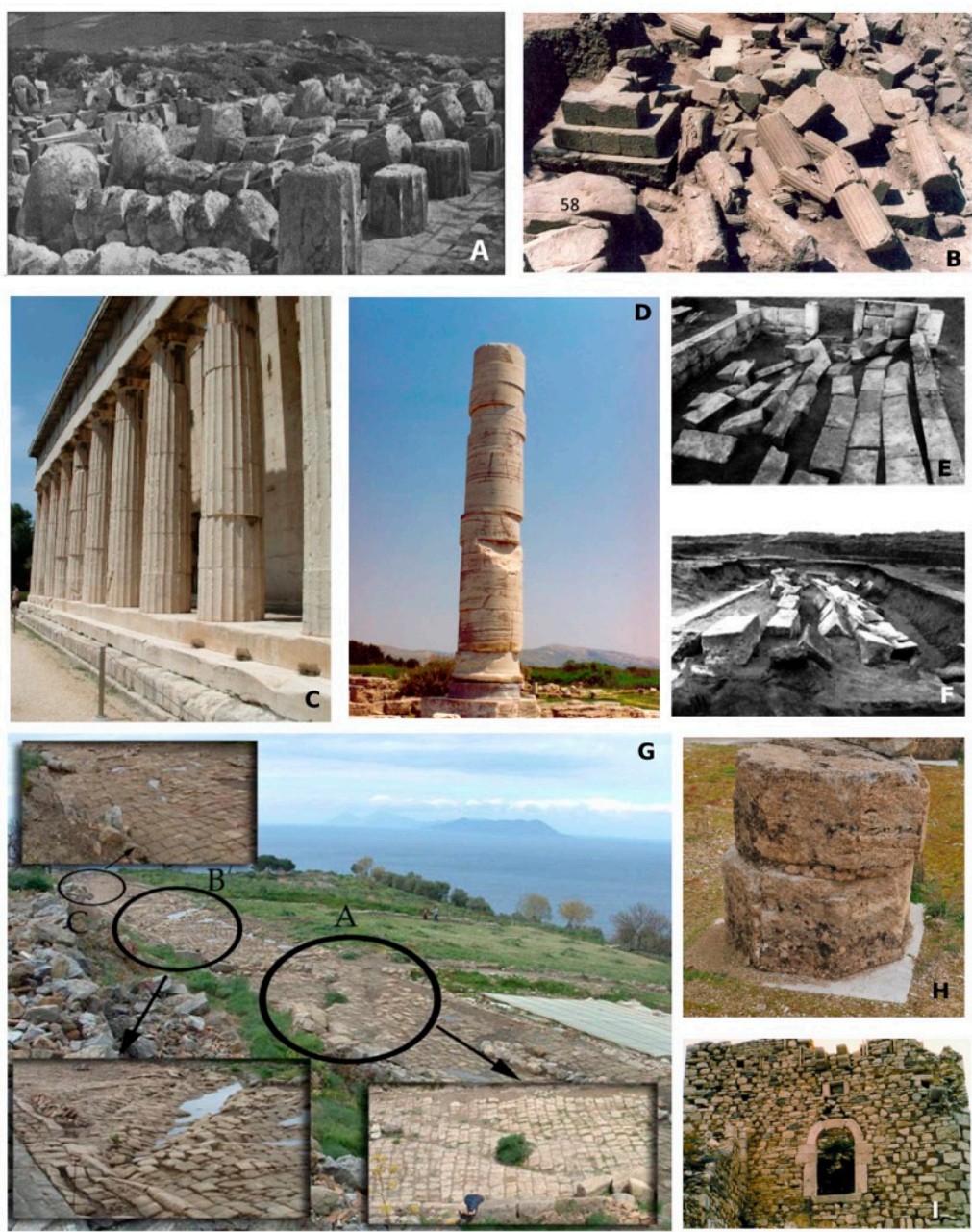

**Figure 1.** Examples of deformation and damage possibly related to earthquakes: (**A**) Oriented collapse in Temple C of Selinunte [26]. (**B**) Toppled columns in the Abakainon necropolis [27]. (**C**) Sinusoidal offset in the columns of the Hephaestus temple in Athens [27]. (**D**) Offset of column drums in the temple of Hera, Samos [27]. (**E**,**F**) Oriented collapse in Triolo Temple [27]. (**G**) Deformed decumanus in Tindari. The letters A, B, and C indicate characteristic deformations of paved floors, likely caused by the amplification of seismic waves [21]. (**H**) Offset in column drums in the Agora of Segesta (photo Bottari). (**I**) Sliding of the keystone in the tower of Logothetis, Samos [27].

## 2. Local Site Effects in Archaeoseismology

### 2.1. Introduction

Quantitative investigations in archaeoseismology have two primary purposes: firstly, the accurate assessment of the ground shaking responsible for causing structural damage, and secondly, the identification of the specific seismic source by analyzing the seismotectonic model [11]. Nonetheless, challenges arise when attempting to correlate the damage observations across adjacent sites and when trying to establish the precise dates for destructive events, primarily due to data limitations and uncertainties within a confined

affected area [28]. These factors can result in distorted estimations of the magnitudes of ancient earthquakes, underscoring the vital necessity for the systematic consideration and evaluation of the localized effects of seismic sites in archaeoseismic research [28].

Ignoring the phenomenon of soil amplification, which involves the intensification of seismic waves when passing through specific soil types, within archaeoseismological studies can result in an overestimation of the magnitudes of ancient destructive earthquakes [11]. To accurately simulate seismic ground motions, it is crucial to include multiple horizontal components as earthquake input signals in site response analyses [29]. Furthermore, conducting on-site measurements of dynamic soil properties, such as density, shear wave velocity, and damping, offers significant advantages [30]. Therefore, the incorporation of quantitative tools from the field of seismic engineering becomes essential for estimating the site-specific local effects in archaeoseismology [8].

Estimating the surface ground motion in archaeoseismic studies can be achieved by using empirical methods, such as analyzing recordings of real earthquakes [28], or numerical techniques like stochastic methods or Green's function, which represents the response of a site to a point seismic source [15,31]. Field tests and analytical/numerical models are employed to evaluate seismic site amplification by studying the dynamic responses of sites using active sources, ambient noise, and real earthquakes [28,32]. Analytical/numerical models play a crucial role in quantitative archaeoseismology, as they help with understanding the characteristics of seismic wave propagation in sedimentary basins when instrumental earthquake records and historical macroseismic intensity data are not available [21,22,33–35]. These models require a well-defined geotechnical model that incorporates the soil layer geometry, dynamic properties of each layer, subsurface incident motions, and synthetic seismic records derived from bedrock sites. Synthetic ground motions are calculated based on carefully selected earthquake source parameters. Incorporating parameters like the extent of fault rupture, dimensions of the faulting area, seismic moment (Mo), and moment magnitude (Mw), which are tied to a seismotectonic model depicting the specific geographic region under investigation. These artificially generated ground motions function as input seismic signals for the computation of site amplification effects and the ensuing surface ground motion unique to the particular site.

Additionally, in archaeoseismic research, it is crucial to consider the environmental connection between archaeological sites and the surrounding landscapes, taking into account archaeological, historical, and geological knowledge. The geological characteristics of the archaeological site, such as whether the settlement developed on sediments/soils deposited on hard rock, directly on hard rock, or in terrain conditions resembling specific geomorphological scenarios such as hills, valleys, slopes, or other topographic features, can significantly influence the behavior of seismic waves and the resulting ground motion [36].

Such geological/geomorphological characteristics described above can increase the seismic shaking impact on archaeological structures, leading to an amplification of the seismic wave, with devastating effects on historical buildings. In such cases, even a low-magnitude local earthquake can cause total collapses under specific conditions (e.g., topography, geology, source parameters). For example, the Santa Venerina earthquake in 2002, located in southern slope Mount Etna, showcased these characteristics, causing significant structural damage despite its relatively low magnitude [37].

Numerous studies in seismic engineering ([11], and references therein) have been conducted to simulate earthquakes on specific historical buildings and estimate earthquake parameters. The findings consistently underscore the pivotal role played by the building's response to even minor changes in the geometry or input parameters of ground motion during the reconstruction of past seismic events.

Consequently, solely simulating seismic events on the building without considering the geological stratigraphy cannot be considered comprehensive or fully accurate. Geological stratigraphy, which involves studying the layers of Earth's materials and their properties, plays a vital role in understanding how seismic waves propagate through the ground. Ignoring this aspect can lead to an incomplete understanding of the seismic behavior,

potentially overlooking critical factors that influence the building's response to earthquakes. Therefore, a holistic approach that considers both the building's characteristics and the geological context is essential for more reliable seismic simulations and risk assessments in archaeoseismology.

*2.2. Site Characterization*

Site characterization is a fundamental aspect of archaeoseismology, playing a key role in understanding the seismic response of archaeological sites to historical or prehistoric earthquakes. This multidisciplinary approach involves a comprehensive analysis of the geological, geotechnical, and geomorphological factors that influence the behavior of seismic waves within the site [35,36]. It entails a meticulous examination of soil types, stratigraphy, rock properties, and topographical features, as well as the utilization of geophysical surveys like seismic refraction, electrical resistivity, and shear wave velocity measurements to assess subsurface properties. By integrating these data, researchers construct a precise 2D model of the site, allowing for the simulation of seismic effects and the identification of potential causative fault sources.

Comprehensive site characterization offers valuable insights into the local geological conditions, which have a substantial impact on the ground shaking amplitude and frequency content. To study the local site effects and identify the causative faults, the integration of geophysical techniques and geochemical data analysis becomes crucial (Figure 2). Archaeoseismologists can combine these methodologies to precisely determine the location of seismic causative sources, particularly in sites where the evidence is unclear [11,15,16]. This, in turn, fosters a deeper understanding of earthquake hazards and their influence on archaeological structures. Nonetheless, achieving precise site characterization is not always feasible. Direct investigations, such as drilling, are often impossible in archaeological contexts. In such cases, indirect investigations, such as geophysical techniques, can serve as trustworthy alternatives. These site characterizations should encompass the delineation of primary geological and geomorphological structures, as well as the determination of shear wave velocity and its subsoil variations [38]. The seismic shear wave velocity model stands as a fundamental piece of information for calculating the seismic response of the site, in conjunction with geotechnical characterizations (considering density and material behavior under dynamic stresses) of the seismic layers.

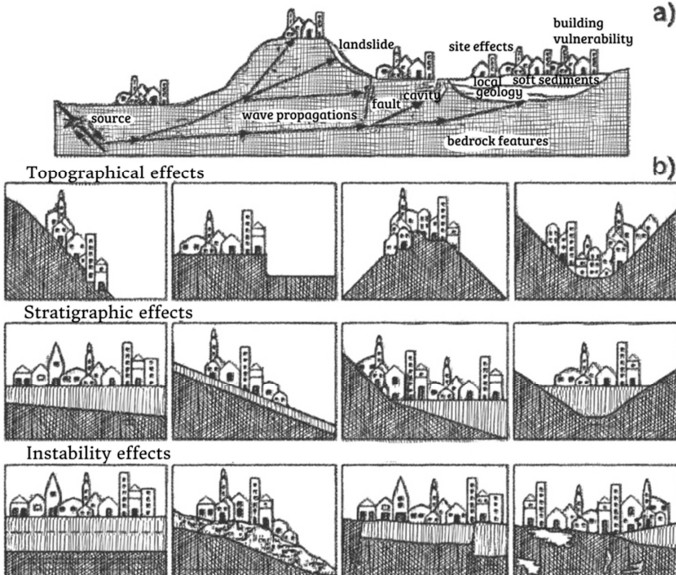

**Figure 2.** (**a**) A sketch illustrating the influences that can affect a seismic signal as it propagates from the source to the free surface of a terrain. (modified from Panzera et al. [39]) (**b**) A scenario depicting local seismic response. (modified from Panzera et al. [39]).

*2.3. Geophysical Techniques*

The Horizontal to Vertical Spectral Ratio (HVSR) and Multichannel Analysis of Surface Waves (MASW) represent advanced geophysical methodologies employed for a thorough assessment of seismic parameters and detailed site-specific seismic response investigations (e.g., [38,40–43]). The HVSR technique [44,45] is a passive seismic method that exploits environmental noise, eliminating the need for an energization system. Notably, the HVSR method has been extensively tested and utilized at various sites by different researchers [46–51]. This noise primarily comprises surface waves, including Rayleigh waves and Love waves, resulting from the constructive interference of P and S waves in the surface layers.

In contrast, MASW employs an array of multiple geophones or seismometers arranged in a linear or grid pattern on the ground's surface [52]. It captures surface waves, including Love and Rayleigh waves, propagating horizontally along the ground. These recordings are subsequently used to construct subsurface S-wave velocity profiles. MASW primarily serves the purpose of subsurface imaging and characterization, providing essential information about the S-wave velocity structure, a critical factor for assessing the soil stability and potential seismic hazards. Generally, the MASW technique does not reach great investigation depths, especially in soils with homogeneous behavior in which the dispersion curve is not created.

Both these methodologies entail specialized data analysis using a network of seismometers and accelerometers. In HVSR, data analysis revolves around spectral analysis to unveil the predominant or natural resonance frequency of the subsurface and provide insights into the depth of specific geological layers and their material properties. In MASW, data analysis focuses on the dispersion curve, which establishes the relationship between wave velocity and frequency and plays a crucial role in the inversion process for obtaining S-wave velocity profiles in the subsurface.

This analytical process comprehensively includes investigating various aspects of the recorded seismic signals, encompassing oscillation frequencies, amplitudes, phase velocities, and dispersion characteristics. The meticulous examination of these complex parameters empowers researchers to gain invaluable insights into seismic events and their multifaceted attributes, transcending mere earthquake magnitude determination. It offers a sophisticated understanding of seismic wave interactions with geological structures and subsurface materials. Consequently, the assessment covers the entirety of seismic hazards, ranging from ground shaking characterizations to the evaluation of their effects on diverse structural configurations.

HVSR and MASW techniques significantly contribute to a precise understanding of the dynamic interplay between seismic waves and subsurface geological compositions. This heightened level of scientific insight proves indispensable for improving seismic hazard assessments, seismic risk management, and the development of resilient structural and infrastructural systems in seismically active regions.

The integration of MASW and HVSR techniques represents a powerful approach to gaining a comprehensive understanding of subsurface seismic properties [53,54]. Combining these methodologies allows researchers and geophysicists to link subsurface geophysical attributes with ground motion characteristics. Such integration proves valuable in seismic hazard assessments, offering a holistic perspective on seismic events, their effects on geological formations, and their potential implications for structural and infrastructural resilience, thereby contributing to more informed decision-making in earthquake-prone regions.

*2.4. Site Amplification*

The amplification of seismic waves, as influenced by local soil and rock conditions, plays a crucial role in shaping the ground motion characteristics at a specific site. These local geological conditions have a profound impact on the interaction of seismic waves with the Earth, thereby affecting both the amplitude and frequency content of ground shaking.

Different soil types exhibit varied responses to incoming seismic waves, with some soils amplifying or attenuating ground motion. To comprehend and quantify these effects, specialized analysis techniques come into play. Among them, site response analysis emerges as a valuable tool for evaluating and characterizing seismic amplification phenomena [55–57]. This method empowers researchers and engineers to gain insights into the intricate interplay between geological factors and seismic ground motion, thus contributing to a more comprehensive understanding of earthquake hazards and risk assessments.

The assessment of site response using a simplified method, which involves estimating the subsoil category, is applicable only under specific stratigraphic site conditions and for the design of simple structures. In cases where the structure to be constructed holds critical public or strategic functions, or if there are potential variations in the vertical shear wave velocity profile or complex topographical features, the numerical approach becomes necessary for calculating the local seismic response. The steps involved in conducting a local seismic response analysis include:

- Selection of a seismic input spectrum compatible with the site's rigid soil category.
- Definition of the seismo-stratigraphic model and determination of the physical-mechanical characteristics of the materials.
- Calculation of the seismic response at the surface.

The choice of compatible accelerograms depends on the underlying seismic hazard and an understanding of the maximum expected earthquake at the site. The construction of the subsoil model is typically achieved through the utilization of geophysical methodologies, but a comprehensive understanding of the area's tectonic model is also essential to understand the limitations of the modeling process.

*2.5. Topographic Effects*

The geological and topographical characteristics of a region play a key role in shaping the effects of seismic activity on the ground. Factors like the presence of hills, slopes, valleys, or other topographic features can substantially influence the propagation of seismic waves, causing complex variations in ground motion [39]. When seismic waves come across topographic irregularities, their behavior can lead to phenomena such as wave focusing, scattering, or interference. As a result, ground shaking intensities can vary significantly across different locations, even within close proximity to the earthquake's epicenter.

To understand the topographic effects on seismic waves, numerical modeling techniques take center stage. Finite element analysis, a well-established method, is often used in earthquake engineering and seismology. This numerical tool enables researchers to simulate the interactions between seismic waves and complex topographies. By modeling the behavior of seismic waves as they navigate through areas with varying elevations and geological properties, researchers can gain valuable insights into the ground motion patterns that arise.

Understanding these topographic influences on seismic waves and ground motion is essential for accurate seismic hazard assessments and risk mitigation [58]. By studying the interactions between topography and seismic activity, researchers can better predict the areas that are most vulnerable to strong ground shaking during earthquakes, aiding in informed urban planning, infrastructure development, and emergency response strategies. This comprehensive understanding of the interplay between topography and seismic waves is an integral component of earthquake research and awareness efforts.

*2.6. Ground Motion*

Ground motion, as experienced in archaeological sites, stands as a main factor shaping the intricate interplay between seismic activity and cultural heritage preservation [28,59]. The ground motion denotes the complex oscillations and vibrations of the Earth's surface triggered by seismic waves during an earthquake. In archaeological settings, this phenomenon carries substantial significance due to its potential to cause structural damage, expose the integrity of historical buildings, and impact the preservation of cul-

tural heritage. Several elements come into play when evaluating ground motion within archaeological contexts.

Foremost, the inherent characteristics of the archaeological site hold considerable sway. The geological, geotechnical, and topographical features of the site dictate the extent and nature of ground motion [33,39,60]. The local geology and soil composition, whether composed of soft, loose soils prone to amplifying seismic waves or hard rock offering some degree of attenuation, significantly influence ground motion. Furthermore, the site's topography, encompassing hills, slopes, valleys, and other features, may lead to wave focusing, scattering, or interference, engendering variations in ground shaking intensity.

The seismic source itself is another critical determinant. Factors such as the depth, magnitude, and proximity of the seismic source to the archaeological site wield substantial influence. Sites in close proximity to active fault lines or seismic sources are inherently more susceptible to intense ground motion. In cases where ground motion interacts with archaeological structures and artifacts, the vulnerability of these cultural assets comes into play. The construction methods, materials, and historical age of these structures play a main role in their susceptibility to seismic impacts. Older, less-reinforced structures are prone to collapse and, therefore, are at higher risk.

Beyond these elements, amplification effects tied to local soil and rock conditions are predominant. Certain materials possess the capacity to significantly amplify ground motion which, in turn, can intensify seismic impacts. This amplification often occurs as a result of site resonance, where the natural frequency of the geological and geotechnical properties of the site matches the frequency of incoming seismic waves [61]. This resonance can intensify the ground motion, thereby increasing the potential for damage. Understanding these resonance dynamics is instrumental in the assessment of vulnerability and the formulation of protective measures.

The study of ground motion within archaeological contexts necessitates a multi-pronged approach encompassing geophysical surveys, comprehensive site characterization, and numerical modeling [28]. It aids in predicting how seismic waves will interact with the site, its structures, and its invaluable artifacts. This predictive capability is indispensable for developing strategies aimed at mitigating the risks and impacts of seismic events, safeguarding historical heritage, and assessing vulnerabilities.

In this intricate interdisciplinary field, archaeoseismologists, geologists, and cultural heritage experts collaborate on data collection and analysis. They collaborate to decode local site effects, assess ground motion parameters, and thus ensure the sturdy protection and preservation of cultural and historical monuments, bridging the gap between Earth science and heritage conservation.

### 2.7. Selinunte's Case Study of the Local Site Effect

Selinunte, located in western Sicily, is a renowned archaeological park celebrated for its rich historical significance, currently constituting one of the largest park in Europe. The landscape is morphologically divided into the Western hill, the Acropolis, and the Eastern hill by two rivers. The proposition attributing temple destruction to seismic events was initially posited by Hulot and Fougères [26]. Recent scholarly investigations by Guidoboni et al. [62], Bottari et al. [63], and Schwellenbach et al. [23] have substantially strengthened this hypothesis through meticulous archaeoseismological studies.

Guidoboni et al. [62] and Bottari et al. [63] utilized a quantitative methodology [7–10] to discern earthquake traces within Selinunte's remains spanning from 400 B.C.E. to 1200 C.E. In contrast, Schwellenbach et al. [23] adopted a qualitative approach. The research [63] followed a quantitative approach, involving a systematic process. Initially, a thorough review of archaeological literature sought to identify destruction layers and earthquake-related deformations. Subsequently, an in-depth examination of sites aimed to identify seismic effects not explicitly documented in the literature, focusing on characteristic de-formations.

Historical scrutiny of sites potentially affected by earthquakes involved a detailed analysis of periods of occupation, wartime activities, and structural evolution. This investigation aimed to differentiate seismic impacts from alternative explanations. To broaden the study's scope, efforts were made to identify evidence of earthquakes in nearby archaeological sites. Dating of earthquake occurrences relied on archaeological artifacts, particularly coins, pottery, and inscriptions, with a meticulous examination of ancient findings. Subsequently, parameters of documented earthquakes, including intensity and causative seismogenic structures, were estimated.

The initial earthquake, occurred between 370 and 300 B.C.E., is associated with the destruction of temples in the Western hill, including Temple R on the Acropolis (Marconi, personal communication). A subsequent earthquake, occurring between 330 and 500 C.E., resulted in the collapse of temples both on the Acropolis and in the Eastern hill [63].

The extent to which seismic damage led to the collapse of all Selinunte temples was a matter of ongoing scholarly debate, primarily due to the absence of definitive seismic evidence [63]. In contrast to the oriented collapse observed in Temple C on the Acropolis and the other temple on the Western hill, temples situated on the Eastern hill demonstrated a chaotic collapse. This pattern of collapse could be ascribed either to destruction caused by human activities or to a seismic sequence, as highlighted by Guidoboni et al. [62]. Their comprehensive study covered a broad range of temple collapses, providing a comprehensive historical context. According to the authors [62], the first earthquake occurred between the 4th and 3rd centuries B.C.E., whereas the second one occurred from the 6th century to the 13th C.E. In contrast, Bottari et al. [63] concentrated on specific temples, providing precise insights into earthquake-induced collapses, with a more focused approach based on combining historical and archaeological data.

The seismic dating of Temple C's destruction is established through the analysis of discovered artifacts, offering both lower and upper bounds. A Christian bronze lamp dating back to the Late Roman Period (330–400 C.E.) serves as a lower bound, while a pottery lamp unearthed in the Roman house, destroyed by fallen columns, aligns with the 4th–5th centuries C.E., indicating the collapse occurred around 350–500 C.E. In summary, the columns of the temple collapsed between 330 and 500 C.E. [63].

Schwellenbach et al. [23] improved the study by conducting a comprehensive examination of the stratigraphy in critical zones of Selinunte, such as the Acropolis, the Eastern hill, and the Western hill. Their investigation into local site effects yielded significant in-sights, confirming earlier geological hypotheses about the presence of a calcarenite layer in the Acropolis and Eastern hill sites. Conversely, the absence of a calcarenite layer in the Western hill site suggests different site effects.

Based on Schwellenbach et al.'s [23] results, it can be suggested that the initial earthquake event (370–300 B.C.E.) had a lower magnitude compared to the subsequent seismic event (330–500 C.E.), leading to the destruction of all temples on the Acropolis and the Eastern hill. However, determining whether this later event was a singular seismic event or part of a larger seismic sequence remains challenging due to the inherent difficulties in studying ancient earthquakes. In conclusion, Guidoboni et al. [62] provided a comprehensive historical overview contextualizing temple collapse, while Bottari offered more precise insights into specific temple collapses [63]. Finally, Schwellenbach et al. [23] made significant contributions to exploring local site effects, further enhancing our understanding of seismic events within Selinunte's historical record.

## 3. Geochemistry: A Tool for Archaeoseismology

Geochemistry plays an essential role in the field of geology, contributing significantly to our understanding of Earth's processes, geological history, and, notably, in the identification of active buried faults [64]. The primary goal of geochemistry is to unravel the complex chemical interactions between Earth's materials, enabling geoscientists to decipher the past, present, and future of our planet. In the geological field, geochemistry serves as a fundamental tool for investigating a wide range of phenomena, from petrology and

mineral exploration to environmental studies and geochronology [65]. Its utility extends to the identification of active buried faults, where it proves invaluable. Geochemists employ various techniques to analyze the composition of minerals, rocks, and fluids, making it possible to detect subtle changes or anomalies associated with fault activity [66]. These alterations include variations in mineral assemblages, isotopic compositions, or the geochemical signature of fluids influenced by fault zones. This information aids in the precise delineation and understanding of hidden fault systems, essential for seismic hazard assessment. Moreover, geochemistry finds application in archaeoseismology, where it is fundamental in identifying active buried faults and its correlation to past seismic events, thereby contributing to seismic risk assessment. In principle, geochemistry's role in geological research demonstrates its critical importance in unraveling the Earth's dynamic processes and mitigating the impact of seismic events on society [67].

Geochemistry, when employed in the field of archaeoseismology, emerges as a potent and multifaceted tool for unraveling the seismic histories surrounded within the records of archaeological sites. This interdisciplinary approach entails a meticulous exploration of the chemical elements and isotopic compositions present in thermal fluid.

The utility of geochemistry in archaeoseismology lies in its ability to unveil subtle yet significant traces of seismic disturbances. Such disturbances may manifest as distinct chemical anomalies, liquefaction features, fault gouges, or alterations in mineralogical compositions within the geological strata or archaeological layers [68]. These refined but revealing signs serve as empirical records of past seismic events that have reverberated through time.

By subjecting gas and fluid samples to rigorous geochemical analysis, archaeoseismologists can access valuable information concerning the proximity and nature of active faults. Such information is of great importance, not only for reconstructing the seismic history but also for understanding the seismic risks that ancient civilizations encountered. Furthermore, it provides a unique perspective for evaluating how seismic events have influenced cultural landscapes and heritage over millennia.

### 3.1. Geochemical Tecniques

Geochemical techniques for the analysis of gases and fluids encompass a variety of methods, mainly with Gas Chromatography and Mass Spectrometry (MS) equipment [68,69]. These techniques allow us to analyze the chemical and isotopic composition of gaseous species and, consequently, recognize the origin present in fluid emissions. Additionally, the use of accumulation chambers is vital in geochemical investigations, as they allow for the controlled collection and monitoring of gases over time, providing insights into the dynamics of subsurface fluid and gas reservoirs. These comprehensive techniques serve as indispensable tools for various purposes, including hydrocarbon exploration, environmental monitoring, and understanding the geological processes that shape our planet. Furthermore, they play a pivotal role in the identification of active buried faults, thus aiding both geological fieldwork and archaeoseismological studies dedicated to identifying and characterizing active faults.

In the field of geochemistry, these techniques find essential applications in the monitoring of gases in thermal sites. By examining the composition and flux of gases like carbon dioxide, sulfur compounds, and noble gases in geothermal areas, geochemists gain valuable insights into the subsurface dynamics of these thermal sites [24,25]. Continuous gas monitoring allows for the detection of changes in gas emissions, which can serve as early indicators of potential volcanic or geothermal activity. Geochemical data play a crucial role in risk assessment and hazard mitigation, contributing to the safety and sustainable utilization of geothermal resources. This monitoring is not only vital for understanding geological processes, but also for ensuring the safety of communities living near thermal sites. Thus, the intersection of geochemistry and gas monitoring serves as an invaluable tool in assessing and managing the environmental and geological aspects of thermal areas (Figure 3).

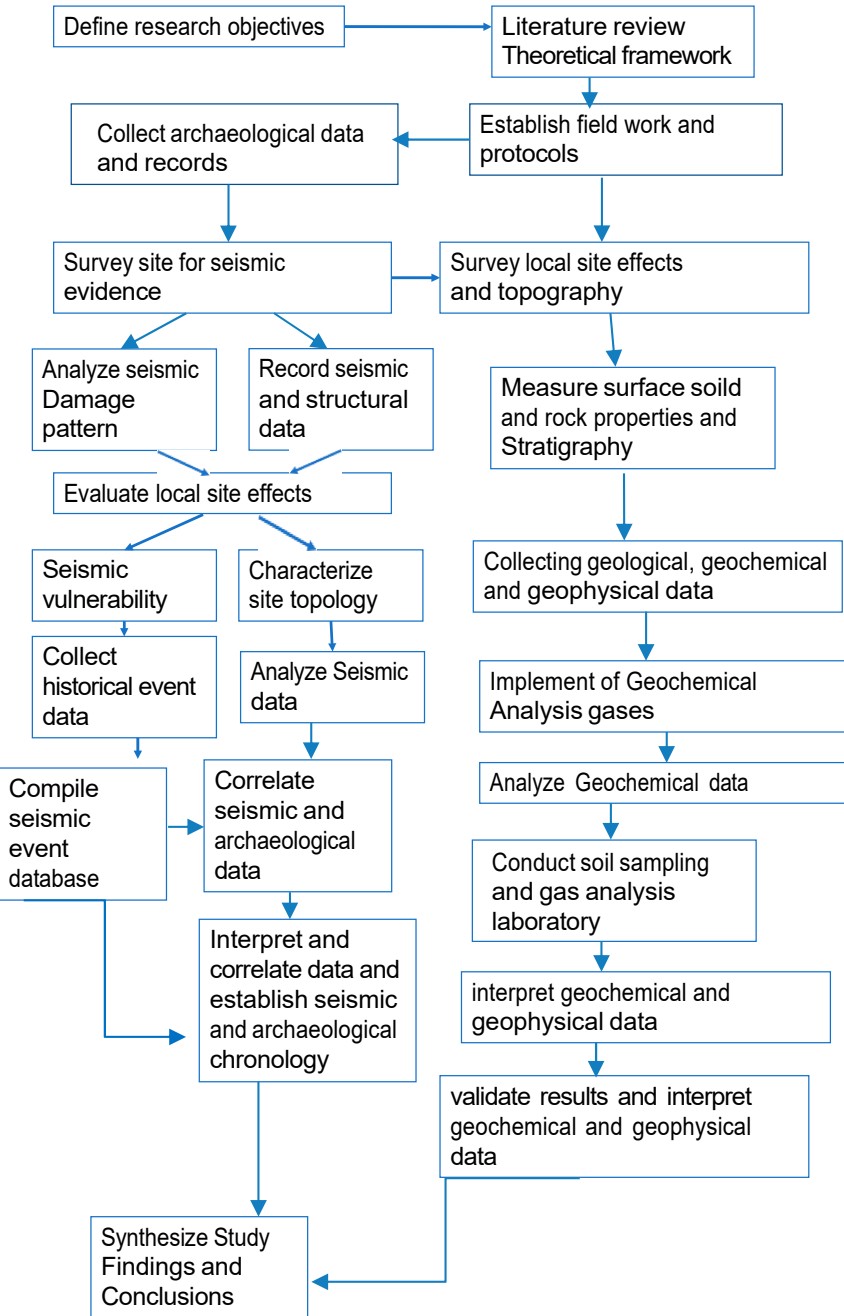

**Figure 3.** Archaeoseismology, local site effects, and geochemistry flowchart.

### 3.2. Santa Venera al Pozzo Case Study

The Santa Venera al Pozzo (SVP) site, located in eastern Sicily near Catania, is renowned for its historical significance as an ancient thermal location [70]. The site, positioned on the lower eastern flank of Mt. Etna volcano near Acireale, in proximity to a major tectonic fault system experienced significant earthquakes in the past, including events in 1865, 1911, and 2002. Recent seismic activity, such as the earthquake on 26 December 2018, occurred during the volcanic eruption from the 24th to the 27th.

The local seismicity is attributed to a combination of regional tectonics, repeated magmatic intrusions, and the lateral spreading of the volcano, evidenced by aseismic creep motion along local faults.

Aseismic creep episodes, causing substantial ground cracking, have been observed in conjunction with seismic crises along other tectonic structures on the eastern flank of the volcano. Within the archaeological site of Santa Venera al Pozzo, the foundations of a

Roman temple and other ancient remains exhibit clear faulting and deformation, notably in a system of fractures offsetting Roman structures (Figure 4). Offset in foundation can be clearly related to an activation of a fault during an earthquake.

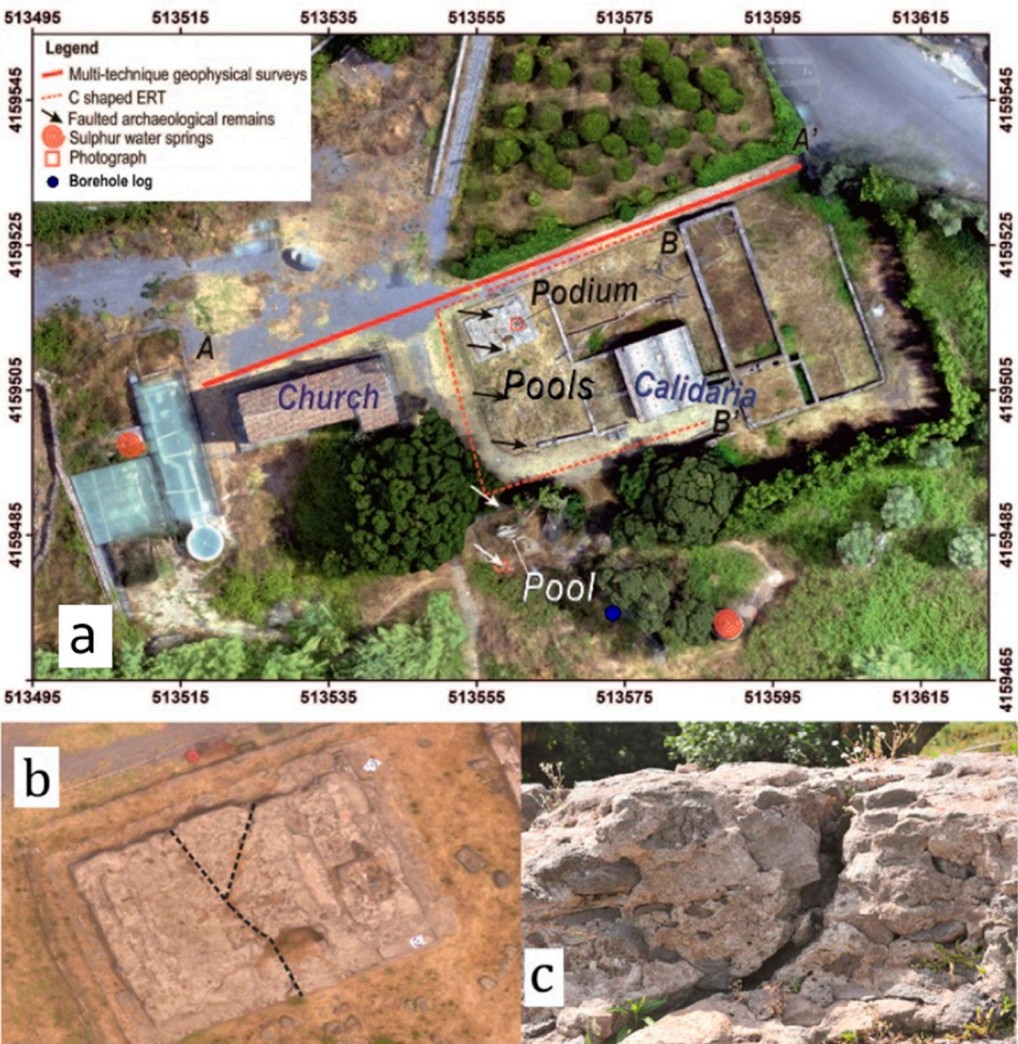

**Figure 4.** (**a**) Digital Surface Model (DSM) obtained by an Unmanned Aerial Vehicle of Santa Venera al Pozzo site (modified from Bottari et al. [24]), arrows indicate ground deformation ascribed to seismic and/or creep activity which extend for about 40 m in a N-S oriented direction; red circles indicate the Sulphur water springs used for sampling gas; (**b**) aerial view of the faulted podium highlighted by black dot line; (**c**) detail of the faulted podium from lateral view.

Furthermore, the presence of thermal water indicates a convective uprise of crustal or subcrustal fluid that receives heat from a deep-seated source. Whatever the fault system driving the fluids to the surface, it is evident that the thermal spring of Santa Venera al Pozzo indicates the occurrence of favorable permeability conditions in the crust that can only exist because of deep structures. The anomalous upflow zone well matches the geometry of the dislocated zone, mainly in a north-south direction, offsetting the foundations of the Roman podium, some pools, and minor walls nearby, based on a multi-techniques geophysical survey recently carried out in the area [71]. However, the geophysical survey did not allow identifying a clear fault plane at depth but rather highlighted a broad anomalous zone interpretable as a fault zone. The lack of a clear discontinuity has initially been ascribed to the shallow depth of investigation, the types of outcropping rocks, and the presence of fluids [71]. However, the collected geochemical data show the existence of different and unknown hidden faults in the site.

Our primary objective was to assess structural damage resulting from the pivotal event of 251 C.E. [71]. To accomplish this, we adopted a multidisciplinary approach that integrated historical, archaeological, geological, geophysical, and seismological data. By effectively excluding human-related causes, we underscored seismic activity, particularly earthquakes, as the most probable cause.

The geological landscape of the region, profoundly influenced by volcano-tectonic processes and deep-seated faults, emerges as a main factor in the site's susceptibility to seismic events. The intricate interplay of geological elements, including thermal springs and deep fault segments, deepens our comprehension of the underlying factors influencing the site's structural evolution.

Furthermore, the geochemical analysis of fluids and gases emitted near the faults at Santa Venera al Pozzo plays an indispensable role, as documented by Ferrara [72], and Bottari et al. [24]. This analysis enabled the interpretation of the chemical composition and isotopic characteristics of the emitted gases.

The natural thermal springs of SVP are characterized by low water temperature (around 22 °C), low water output (around 2 ls), and a distinct smell of $H_2S$. In some areas of the ancient thermal baths, intermittent emissions of free bubbling gases can be observed from the water surface in the thermal springs.

Previous studies hypothesize that the origin of SVP waters is primarily from rain, with a minor percentage coming from seawater intrusion. However, the elevated B/Cl and Li/Na ratios in Santa Venera al Pozzo springs, significantly higher than those of seawater, indicate marked water/rock interaction.

Grassa et al. hypothesized a seawater contribution to the aquifer feeding the SVP springs based on isotopic data [73]. They also found reduced conditions within the local aquifer due to methane (and $H_2S$) presence in the dissolved gas phase. According to some authors [74,75], the SVP water may originate from a thermal reservoir whose fluids result from mixing between groundwater and thermal brines from sedimentary layers, similar to those emitted from mud volcanoes located on the lower SW slopes of Mount Etna [76]. Strontium isotopes, as calculated by Liotta et al. [75], suggest that about 10% of deep fluids are involved in the mixing process at SVP.

The chemical composition of SVP gases is characterized by high concentrations of methane ($CH_4$) and carbon dioxide ($CO_2$), and minor contents of ethane ($C_2H_6$), sulfuric acid ($H_2S$), and carbon monoxide (CO). Nitrogen ($N_2$) and oxygen ($O_2$) are also present, but with a $N_2/O_2$ ratio of 11.09, significantly higher than that of air (3.75). This high ratio results from the consumption of reduced gas species such as $H_2S$ within the shallow hydrothermal system [77].

The isotopic composition of the emitted gases suggests a non-volcanic origin, but a possible association with a deep tectonic system. According to Dongarra and Hauser [77], the isotopic composition of sulfur of sulfates, native sulfur, and $H_2S$ is attributed to processes of bacterial reduction of sulfate, either from evaporitic rocks or seawater, to $H_2S$ and its subsequent re-oxidation in a shallow environment to colloidal sulfur and back to sulfate. Additionally, methane isotopes indicate a combination of microbial generation and thermogenic hydrocarbon sources [77,78].

The carbon isotopes of $CO^2$ suggest a biogenic origin that may have undergone partial isotopic alteration due to interaction with the carbon of methane. The air-corrected isotopic ratio of helium (He) (R/Ra) excludes any mantle-derived contribution of this gas and suggests a crustal origin.

Consequently, it is possible to identify that the fluids emitted at the SVP site result from two components, one more superficial and one deeper.

This geochemical data also played a crucial role in distinguishing between fault types and environments, greatly contributing to our comprehensive understanding of the processes underlying fault activity [24,79]. The unique geochemical signatures associated with fault activity were central in characterizing and identifying the fault responsible for seismic events.

The discovery of significant anomalies in soil $CO_2$, notably coinciding with the predominant trend of hidden faults characterized by a NE-SW orientation, emphasizes the critical role of the geological context in shaping the architectural history of Santa Venera al Pozzo. This further underscores the profound synergy between geological and archaeological aspects.

To gain deeper insights into the geodynamics of the study area, specifically to identify hidden active faults, we conducted a comprehensive geochemical analysis of gases and fluids present at the site in two phases. Initially, we conducted surveys of soil $CO_2$ emissions using the accumulation chamber method in the archaeological area and its vicinity over a two-year period to detect $CO_2$ emissions on a large scale along known or hidden faults [24]. Subsequently, from December 2017 to April 2019, we performed continuous monitoring of gas emissions ($CO_2$, $CH_4$, and $H_2S$) through a microG-C at the site along with a water temperature survey at the Santa Venera al Pozzo thermal springs [25].

The average composition of gases emitted from the SVP thermal waters primarily consisted of approximately 7% $CO_2$ and 15% $CH_4$, with traces of $H_2S$. Water temperature remained relatively stable, varying between 20.8 °C and 23.4 °C, indicating consistent thermal conditions in the aquifer feeding the SVP springs.

High-frequency monitoring of gas emissions and water temperature from the SVP site, with a high degree of precision and accuracy, revealed interesting connections between the transfer of geothermal fluids to the surface and changes in local crustal stress due to tectonic or volcano-tectonic activity.

During the period of monitoring, marked variations in the composition of $CH_4$ and $H_2S$ were observed, while those of $CO_2$ were primarily attributable to seasonal fluctuations. The presence of $H_2S$ was particularly noteworthy, as it coincided with significant seismic events in the Etna area. This suggests that tectonic and volcanic activities have a notable influence on the mixing and release of deep geothermal fluids, which, in turn, impact gas emissions at the SVP site. Notably, there were spike-like emissions of $H_2S$ and $CH_4$ recorded on at least five occasions in December 2018, including the 3rd, between the 6th and the 7th, on the 23rd, and the 26th. During these events, $CH_4$ concentrations exceeded 20%, and $H_2S$ concentrations reached up to 993 ppm. $CO_2$ values remained elevated but relatively stable (see Figure 5) [25].

Due to the organic origin of $CO_2$, we can rule out a direct volcanic influence on the observed changes in $CO_2$ gas concentration at SVP. However, an indirect effect from volcanic activity remains possible. In our case, the massive upward migration of magma inside the shallow portions of the volcano preceding the eruption between December 24th and 27th, 2018 [80–82], may have induced marked increases in large-scale ground permeability, as observed from ground deformation data [80]. This, in turn, caused a more significant release of pressurized gas from the thermal aquifer.

A model of the geothermal system that feeds the SVP thermal springs was developed, depicting it as a reservoir of fluids nearly saturated in reduced gases ($CH_4$ and $H_2S$). This reservoir is highly sensitive to tectonic stresses acting upon it. When tectonic forces alter the aquifer's permeability, the geothermal reservoir becomes saturated with deep gases ($CH_4$ and $H_2S$), effectively acting as a pressure valve that releases these gases based on the rate and intensity of crustal stress.

In this scenario, the relative proportions between $CH_4$ and $H_2S$ depend on the stress rate and its influence on the differential solubility of the two gases in water. Volcanic activity does not appear to directly alter the chemical composition of SVP gases, but it likely induces tectonic movements at the scale of the volcano, which, in turn, influence the chemistry of emitted gases.

This study underscores the significance of multidisciplinary investigations and highlights the valuable insights that geochemical analysis of fluids and gases can provide for archaeoseismological research.

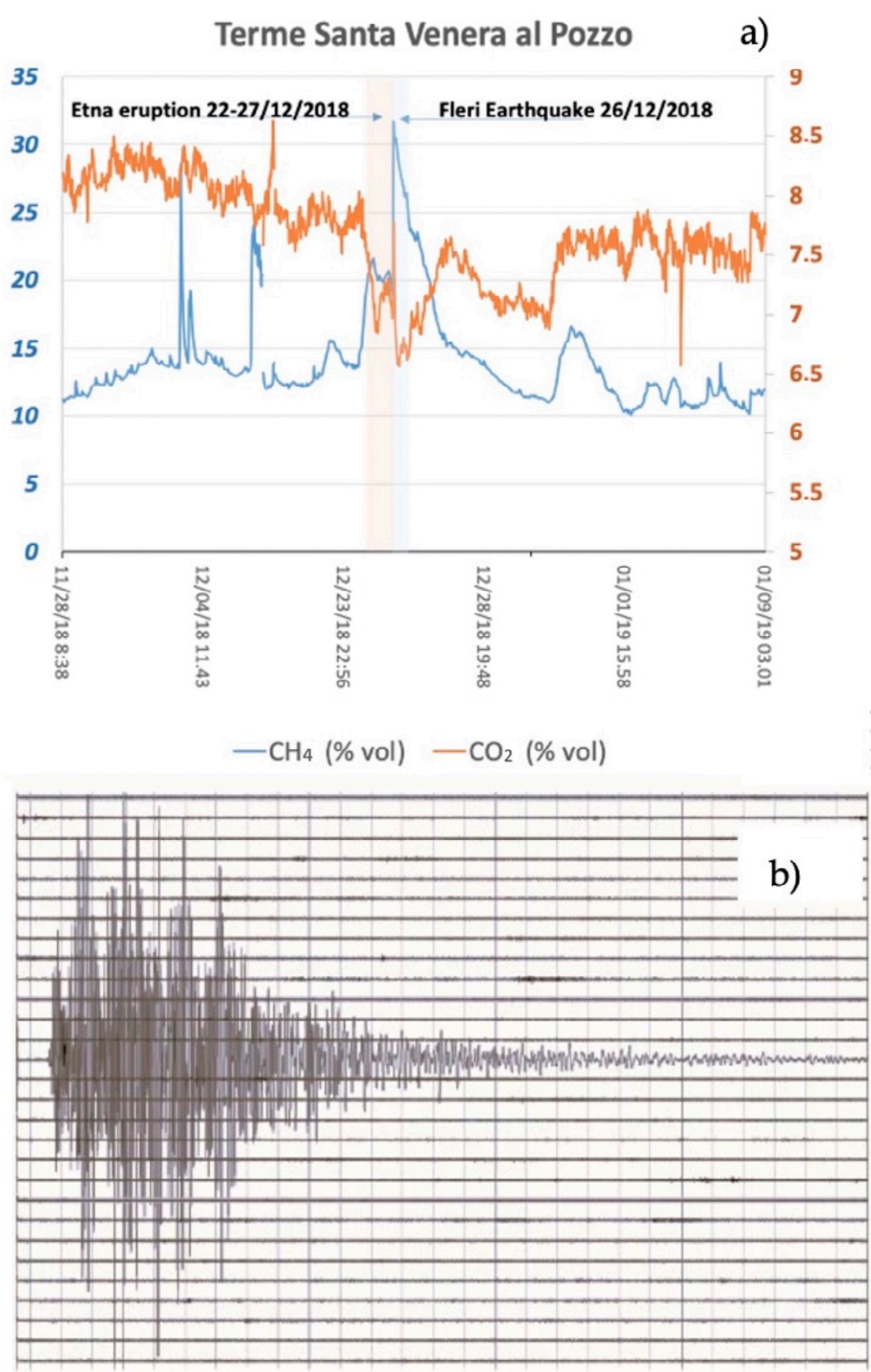

**Figure 5.** (**a**) Temporal variations in $CO_2$ (blue line) and $CH_4$ (orange line) concentrations at the Santa Venera al Pozzo site during the period from 28 November 2018 to 9 January 2019; (**b**) seismogram of the Fleri earthquake recorded by the seismic network of Osservatorio Etneo (INGV).

## 4. Discussion

Archaeoseismology, a methodological approach focused on the examination of past earthquakes through archaeological evidence, presents both strengths and limitations. Let us delve into these advantages and disadvantages.

Strengths: (1) direct evidence: Archaeoseismology yields direct evidence of historical earthquakes by analyzing earthquake-induced damage within archaeological sites. This

evidence encompasses observations of collapsed structures, displaced objects, and various visible indicators of seismic activity, thereby facilitating a more precise understanding of past earthquakes and their repercussions [1–5,7]; (2) localized studies: Archaeoseismology facilitates meticulous, site-specific investigations into seismic activity within distinct archaeological sites or regions. This specialized approach yields comprehensive insights into the seismic history of specific areas, encompassing recurrence intervals and the correlation of earthquake intensities, frequently assessed through the Modified Mercalli Intensity (MMI) scale [16]. It is important to acknowledge, however, that the MMI scale, originally designed for assessing shaking intensity in historical structures, may not be directly applicable to archaeological sites due to variations in construction materials and techniques, particularly in cases where concrete was not used. Nonetheless, the study of shaking intensity in historical structures and its association with Peak Ground Acceleration (PGA) values may offer valuable insights [11]; (3) long-term perspective: Archaeoseismology unfolds a long-term perspective on seismic activity by examining events predating the era of historical records. This extended view unveils information about seismic events that might otherwise remain undocumented or poorly recorded, enriching our overall understanding of earthquake history; (4) combined approach: Archaeoseismology complements other scientific disciplines, including geology and historical seismology, by offering supplementary data and diverse perspectives [8,11,24]. Its role is crucial in addressing gaps within historical seismic catalogs and in verifying or enhancing existing seismic historical records. Furthermore, it can also incorporate local site effects and geochemical analysis, as shown in the flowchart represented in [11,15,25]. As previously mentioned, local site effects provide insights into variations in ground shaking due to geological conditions specific to a given site. This consideration enhances the precision of seismic assessments within archaeological areas by acknowledging that ground shaking may vary significantly from one location to another, even during the same earthquake event. On the other hand, when geochemical analysis is integrated into archaeoseismological studies, it further enriches the interdisciplinary approach. By examining the composition of gases and fluids, it becomes possible to detect indications of active faults concealed in the vicinity of archaeological sites. This geochemical aspect contributes valuable information for understanding the seismic history of an area and evaluating its seismic hazard [25].

Weaknesses: (1) interpretation challenges: The interpretation of earthquake-induced damage in archaeological contexts represents a multifaceted and intricate task. Distinguishing damage triggered by seismic events from that attributed to other factors, such as war destructions, demands meticulous and specialized analysis that can be inherently subjective and, as a result, challenging. The potential for misinterpretation or subjective judgments can significantly affect the precision and accuracy of archaeological earthquake reconstructions; (2) incomplete preservation: Archaeological sites and the structures within them are susceptible to various factors that may result in incomplete preservation. This incompleteness poses a substantial challenge in both identifying and interpreting damage related to seismic events. Factors contributing to incomplete preservation include natural deterioration, human activities over time, and prior archaeological excavations, all of which can complicate the preservation and recognition of seismic evidence; (3) uncertainty in dating: Accurately dating seismic events based solely on archaeological evidence is a difficult task. Establishing a robust chronology of earthquakes often necessitates correlation between archaeological findings and other dating methods, such as radiocarbon or optically stimulated luminescence (OSL) dating, in addition to archaeological data and historical records. Without the incorporation of these supplementary dating techniques, the timing of seismic events may remain shrouded in uncertainty, thus affecting our understanding of their historical context; (4) challenges in estimating earthquake magnitude in archaeoseismology: This difficulty stems from our limited understanding of the proximity of archaeological sites to seismic sources and the intensity of site effects. The absence of precise information on these factors hinders the accurate assessment of seismic event magnitudes in archaeological studies. Additionally, it is crucial to emphasize that, in many instances, identifying

the source fault using solely archaeoseismological methods is unachievable; (5) limited spatial coverage: One of the inherent limitations of archaeoseismological studies is their spatial coverage, constrained to areas with accessible and well-preserved archaeological sites. In many regions, the availability of such well-preserved archaeological remains is limited, making it challenging to achieve a comprehensive understanding of seismic activity on a larger geographical scale. This restriction impacts not only the spatial extent of archaeoseismological findings but also their generalizability to broader contexts.

In conclusion, archaeoseismology, despite its invaluable role in unraveling the historical records of past earthquakes, is not exempt from certain limitations and inherent challenges, demanding careful consideration. The application of rigorous and standardized methodologies, the integration of various lines of evidence, and meticulous acknowledgment of uncertainties in interpreting archaeological data collectively contribute to enhancing the reliability and significance of archaeoseismological investigations.

## 5. Conclusions and Future Perspective

Archaeoseismology is an interdisciplinary field that integrates archaeology, seismology, and geology to uncover the secrets of past earthquakes and their consequences on archaeological sites. Systematic analysis of archaeological remnants, meticulous documentation of damage patterns, and the careful interpretation of geological datasets collectively serve to enrich our comprehension of seismic history. This, in turn, aids in the fine-tuning of seismic hazard assessments and a deeper insight into the complex interplay between seismic events and human societies. The strength of the archaeoseismological approach stems from its unique capacity to provide direct empirical evidence, its long-term temporal perspective, its complementary value to other scientific domains, and its localization of focus. Yet, it does not remain impervious to its own set of challenges—complexities arising from incomplete preservation, uncertainties in dating techniques, constraints on spatial coverage, and occasional interpretative ambiguities.

As we peer into the future of archaeoseismology, it becomes apparent that a wealth of opportunities expects exploration. The following are potential domains of research and advancement.

Interdisciplinary synergies: To support the rigor of archaeoseismological findings, it is imperative that the discipline be more profoundly entangled with other scientific fields. Collaborative interactions with remote sensing, geophysics, geochemistry, and paleoseismology promise to enrich the array of evidence and data sources, thereby enhancing the strength and significance of archaeoseismological inferences.

Quantitative analytics: The objectivity and reproducibility of archaeoseismological undertakings can benefit remarkably from the adoption of quantitative techniques and models for interpreting archaeological data. The utilization of statistical methodologies, GIS-based analyses, and numerical modeling holds the potential to unveil hidden patterns, mitigate uncertainties, and provide more robust explanations.

Regional and global syntheses: A broader comprehension of seismic activity patterns and more comprehensive hazard assessments can be unlocked through the combined pattern and analysis of archaeoseismological data at regional scales integrated with historical ones. Collaborative initiatives and the sharing of datasets across different regions can pave the way for cross-regional investigations and the identification of broader seismic trends.

Public awareness and risk mitigation: Disseminating the insights acquired through archaeoseismological investigations to the wider public domain is of principal importance. Such efforts contribute to enhancing public awareness regarding the historical seismicity of specific regions and, by extension, the potential risks that lie ahead. Informed decision-making with respect to urban planning, infrastructure development, and emergency preparedness can result from this heightened awareness.

In summation, the path of archaeoseismology unfolds in the directions of augmented multidisciplinary engagement, more rigorous quantitative approaches, comprehensive regional and global analyses, and heightened public awareness. These progressive strides

are poised to furnish a more profound comprehension of historical seismic events, their impacts through human societies (i.e., resilience), and a better knowledge of the future seismic hazards.

**Author Contributions:** Conceptualization of methodology based on archaeoseismology with local site effect and geochemical data integration, C.B.; methodology on local site effect, P.C., methodology on geochemistry and data analysis integration F.S.; writing of the original draft preparation, C.B.; writing—review and editing, C.B., P.C. and F.S. All authors have read and agreed to the published version of the manuscript.

**Funding:** This research received no external funding.

**Data Availability Statement:** No new data were created or analyzed in this study.

**Acknowledgments:** The design concept of the work has benefited from discussions with Pierfrancesco Burrato (INGV) and Luigi Ferranti (University of Federico II, Napoli), who I would like to thank.

**Conflicts of Interest:** The authors declare no conflicts of interest.

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
