# Peer review of "Unraveling the Seismic Source in Archaeoseismology: A Combined Approach on Local Site Effects and Geochemical Data Integration"

_heritage, doi:10.3390/heritage7010021_

Round 1

Reviewer 1 Report

Comments and Suggestions for Authors

In my review of the manuscript titled "Unraveling the Seismic Source in Archaeoseismology: a combined approach on local site effects and geochemical data integration," I observed several points that merit attention. Firstly, as this is a review article, I recommend the inclusion of even more references to enhance the paper's depth and credibility. The abstract, while providing an overview of the field, lacks a clear description of the paper itself. I suggest incorporating sentences that succinctly summarize the paper's contribution and significance. It is also crucial to introduce the study in the Introduction section, outlining what readers can expect. The term "interdisciplinary" instead of "multidisciplinary" is proposed as a more fitting descriptor in certain instances, underlining the integration of evidence from various disciplines. The weaknesses of archeoseismology in estimating the magnitude of an event and the limitations in identifying the source fault should be noted in Discussion. I attach the file with specific comments. Overall, I recommend the paper to be published with these minor corrections.

Comments on the Quality of English Language

The English is good. Minor grammatical corrections, such as capitalizing "Earth" and revising word order in specific sentences, are needed for clarity. Additionally, a sentence repetition on page 5 should be addressed, and revise the chapter numbering on page 11.

Author Response

We appreciate your suggestions and comments that have enhanced the paper. We agree with your recommendation to include more references to deepen the paper's content. We have added an outline in the abstract and introduction, and we included magnitude as one of the weaknesses in archaeoseismology. We have accepted all minor improvements to the text.

Thank you for your valuable feedback.

Reviewer 2 Report

Comments and Suggestions for Authors

The aim of this paper is focusing on the combination of local seismic site analysis and geochemical data integration, which results in the unravelling of the seismic source in archaeoseismology. In particular, this approach shows limited information, related to past seismic events and the way they affected archaeological sites.

In my opinion, there are three crucial questions, which are not answered. In particular:

·         The approach described in the paper, is theoretical and it does not correspond to reality, as it cannot be implemented. Therefore, the basis of the entire paper is vague.

·         Although the paper of Ambraseys (2005) is cited, the most significant paper of the same author has not been considered (Ambraseys et al., 2002. Historical Seismicity and Tectonics: The Case of the Eastern Mediterranean and the Middle East, International Geophysics, 81, 747-763). A detailed section with the new contribution of this paper to the archaeoseismology field should have been included.

·         A detailed description of the different seismic catalogues and seismic events is missing, while it is not sufficiently explained how the above are incorporated into the proposed flowchart.

Based on all above, I unfortunately have to reject the paper.

Author Response

We extend our sincere appreciation to referee 2 for the valuable suggestions. We engage in a discussion regarding the accuracy of reviewer 2's assessment of the implementation of our approach. We have presented two distinct case studies: one focused on local seismic effects (Selinunte) and the other one on geochemical survey (Santa Venera al Pozzo). The integration of geochemistry survey in geology is a standard practice. Identifying fault sources solely through archaeoseismology poses challenges, especially in regions like Sicily, where faults are frequently covered by sediments.

Our paper builds upon an already tested approach (Galadini et al.), and we have enriched it with a geochemical survey. The reviewer's opinion on the Ambraseys source seems subjective. Making modifications to the article based on the suggestions of reviewer 2 would fundamentally change its nature, particularly with the addition of a debate about seismic catalogs and seismic events. While refraining from delving into the merits of referee 2's decision, we find the reasons for rejecting the article lacking a detailed explanation.

Seismic catalogs in Italy can be considered comprehensive only for the last five centuries and with M > 6. For the period preceding this, the scarcity of historical sources is substantial, especially for the first millennium. This is an era where archaeology can offer numerous traces, with abandoned settlements preserving evidence of earthquakes through collapses. The disagreement appears to revolve around the use of archaeoseismological methodology, which is not recognized by many researchers as a discipline for implementing historical seismicity.

We appreciate your valuable feedback.

Reviewer 3 Report

Comments and Suggestions for Authors

This is more or less a state-of-the-art paper on how geochemical data could be incorporated in studies of local site effects to provide the magnitude and level of damage caused by past earthquakes, as well as the location of the seismic source. The paper is interesting and well written and could be published with minor modifications as proposed in the following.
Specific comments:
1. Provide a better-quality photo for fig 1(A), if possible.
2. Insert the caption “2.1 Introduction” directly after par. 2. Consequently change the numbering from 2.1 - 2.5 to 2.2 – 2.6.
3. Par. 2.1.1 is meaningless without being followed by 2.1.2. Merge it therefore into 2.2 (new, as above).
4. It is unclear how the contents of par.2.5 fit within the title of chapter 2 “Local site effects …”
5. Provide a better-quality photo for fig 4(b).

Author Response

We express our gratitude to referee 3 for the valuable suggestions. We appreciate the reviewer's input and have incorporated all the requested changes into the text. Additionally, we have included higher-quality photos (1a and 4b). The paragraph 2.5 addresses the local site response at the Selinunte site, and we have chosen to insert this paragraph after the theoretical part.

Thank you for your valuable feedback.
